# How have urban housing preferences developed in response to the COVID-19 pandemic? A case study of Vienna

**Fabian Braesemann** [1,2,3]*, **Jan Kluge**[4], **Hanno Lorenz**[4]

**1** Oxford Internet Institute, University of Oxford, Oxford, United Kingdom, **2** Einstein Center Digital Future (ECDF), Berlin, Germany, **3** DWG Data Science Company, Berlin, Germany, **4** Agenda Austria, Vienna, Austria

* fabian.braesemann@oii.ox.ac.uk

**Data availability statement:** The anonymised data to replicate the findings of the study are available here: https://github.com/Braesemann/ViennaRealEstate.

## Abstract

The digitisation of economic activity in response to the COVID-19 pandemic has reshaped housing demands, potentially leading to the hollowing out of inner cities due to reduced reliance on traditional office spaces. Vienna, renowned for its progressive and diverse housing structures, offers a compelling case study to dissect the interplay between digitisation and flexibilisation of work trends and local housing preference dynamics. In this study, we investigated a data set of more than 120,000 apartment listings from a large online real estate platform to unveil broader patterns shaping residential preferences caused by the COVID-19 pandemic. In analysing rent premiums, we find that amenities related to work-from-home routines have risen in importance while others remained stagnant or saw declining rent premiums. These findings point towards shifted housing preferences to accommodate the need to work from home. This trend is likely to influence housing preferences long after the end of the pandemic, which demands adjusted city planning and housing policies to build sustainable urban environments for a future of hybrid working.

## Introduction

The new ways of working in the digital era are reshaping ongoing urbanisation trends. While economic, social and cultural life has already increasingly concentrated in urban areas [1], the Covid-19 pandemic might have acted as a catalyst, aggravating and deepening those dynamics. When more and more people work from home, not only will the demand for office space drop, but people's housing preferences might change accordingly. The demand for residential property has already changed significantly [2,3]. At the same time, office buildings see less and less demand [4]. As many employees and companies were forced to upgrade their teleworking capacities during the pandemic, a large number of studies have already registered changes in the demand for (and, hence, the pricing of) commercial property in cities around the world [5–8].

By extension, one could deduce that residential property nowadays combines both home and workplace needs [9]. This stipulates the theoretical argument that due to longer periods

**Funding:** The author(s) received no specific funding for this work.

**Competing interests:** The authors have declared that no competing interests exist.

spent in their own four walls, the property's attributes are an even more important factor in people's housing considerations than they already were before the pandemic. We thus argue for a change in housing preferences: Outdoor areas such as balconies or terraces might have become more important for at-home recreation, while other amenities might have reduced in importance, such as public transport accessibility and parking possibilities. It seems likely that employees will be prepared to pay more for apartments that allow the installation of proper working spaces but accept lower premiums for nearby public transport as they travel to their workplaces less frequently. Hence, a shuffle in urban real estate markets might be ahead and already underway.

To investigate this phenomenon, we focus on the case study of Vienna. The residential property market in the Austrian capital was not spared by the dynamics induced by the pandemic. Vienna is known for its affordable council flats that have contributed to making the city one of the most liveable in the world, as well as for its diverse landscape of districts ranging from a densely populated historical inner city to continuously expanding suburbs. The significance of studying the evolution of housing preferences holds profound implications for urban planning, social policy, and economic development. Vienna serves as a compelling case study, offering a microcosm of diverse housing structures and urban dynamics, serving as a magnifier of the larger trends related to hybrid and remote work patterns affecting cities across the globe.

But how exactly has the pandemic changed the demand for housing in the city? How have rents developed in different parts of the city, and which characteristics of flats have become more important? How can investors respond to the change in demand to create the housing that the population wants in the future? In this paper, we present a quantitative analysis using current data from a large real estate platform in Austria. Given that landlords are able to anticipate and price housing preferences correctly, the data gives insight into tenants' revealed preferences. We compare them before and after the pandemic to understand how Vienna's housing market has changed.

We estimate hedonic price models to find how the premiums of relevant dwelling features have changed between 2018 and 2021/22. We expect to find that the increasing prevalence of telecommuting leads to a significant shift in housing preferences. Specifically, there will be a heightened demand for residences that cater to remote work needs and provide greater flexibility within urban areas. This shift is hypothesized to be driven by the emphasis on property features, such as more open space, that enhance the quality of living and, at the same time, reflect the preference for working from home. Conversely, other features, like access to public transport, are anticipated to diminish in importance, mirroring reduced reliance on commuting to traditional office spaces.

Using data for 120,000 rental apartments advertised on a popular local real estate listing online platform, we indeed find that landlords now seem to anticipate a higher willingness to pay for features related to the quality of living, such as an additional room or an outdoor area like a garden or a balcony. At the same time, they charge less for accessibility to public transport. We also find that the results are highly non-linear and vary between inner and outer districts.

The next section reviews the international empirical literature on the determinants of residential rental prices and how they might have changed during the Covid-19 pandemic. Afterwards, we briefly describe our data and method in before we present our results. The last section discusses the implications of the findings for research and practice, and it lines out avenues for future research.

## Literature review

Covid-19 has left its marks not only in economic data series and national budgets but also in the values that consumers seem to attach to particular items in a post-Covid-19 world. The international literature reports measurable behavioural changes in consumption patterns [10], in stockpiling preferences [11] or in travellers' decision-making [12]. Most authors interpret their results with caution as it is still too early to preclude that those observations are transitory or only expedite trends that had started long before the pandemic. However, while consumers might quickly forget about Covid-19 and adjust their behaviour back to normal in many respects, some markets seem likely to undergo permanent or at least long-lasting change. The housing market could be one of the latter.

How residential rental prices and property values have developed before and after Covid-19 must necessarily reflect two things: First, the respective macroeconomic conditions, including the scope of government aid given to households during the pandemic. And second, possible shifts in tenants' willingness to pay for particular housing features.

While this study focuses on the latter, the empirical literature on the macroeconomic impacts informs the wider context of housing research. Regarding the short-term macroeconomic impacts, particularly for the commercial property sector, [13] found a significant decline in rents during the first half of 2020 in the Asia-Pacific region, particularly in those regions hit hardest by the pandemic controlling for property heterogeneity. These findings, jointly with a decline in office properties, point towards the hollowing out of inner cities [4]. Similarly, Allen-Coghlan and McQuinn [14] find the decline of Irish housing prices generally attributable to the declining mortgage credit market, mainly due to diminishing disposable income and fewer qualifications for mortgages. Employing quantile regressions for Turkish property indices, Kartal et al. [15] find that the pandemic negatively affected property prices overall. Mehta et al. [16] find that in India, not the outbreak but rather the peak of the pandemic correlated strongly with the decline in housing prices. For the housing market in China, Qian et al. [17] analyse transaction-level data from online real estate agents' advertisements and find that the negative impact of confirmed Covid-19 cases on housing prices persisted over time, suggesting long-term impacts with deepening trends; this might be particularly affected by the harsh lockdowns enacted on a community basis linked to the number of confirmed cases. In the same vein, Li and Kao [18] point to the importance of considering the spatial distribution of pandemic impacts on real estate markets. Their findings focus on the heterogeneity across counties, stressing that changes in housing prices are heavily localised. Interestingly, Hoesli and Malle [19] analyse the development of housing prices in Europe before and during the pandemic and find a rising price trend for residential properties.

These insights support the argument that macroeconomic trends set the stage for varying localised changes in housing preferences. As we conduct a case study for Vienna, our results must be interpreted against the background of the Austrian macroeconomic conditions. Here, house prices have started decreasing by the end of 2022. This effect is more pronounced for existing buildings; the prices for newly-built houses are still affected by high construction costs [20].

As households are adapting to novel work and living patterns in the post-pandemic world, preferences and rent premiums are shifting. Such shifts can be related both to apartments' characteristics as well as to their respective location. The literature is clear on the latter: Several studies conclude that households—as a consequence of the pandemic—decisively move away from density. [2,3,21–23] find that the post-pandemic housing demand is shifting further away from the city centres and into the suburbs. As a consequence, there are decreased premiums for accessibility to public transport and connectivity to city centres [8,24,25]. Such

locational issues are closely linked to socio-demographic aspects. Duca et al. [26] shows that rents in metropolitan areas in the U.S. have developed unevenly during the first months of the pandemic. While rents dropped in Black, Latino, and diverse neighbourhoods, they kept increasing in mostly White neighbourhoods.

While the effects of Covid-19 on locational issues are well understood, there is less literature concerning housing features. Duca et al. [27] and D'Lima et al. [28] find that households in the U.S. might now be prepared to pay more per unit of area in larger houses. This might hint towards the fact that households now have a taste for larger homes to install proper working areas, but might also be a mere income effect as households have moved away from the expensive city centres and can now afford larger houses. Chen and Luo [29] observes the same effect for the Chinese city of Shanghai, where it, however, might be driven by the experiences households made during shutdowns rather than income effects. De Toro et al. [30] conclude that due to the increased time spent at home, features such as common areas, access to more spacious areas such as parks and other home features, like natural lighting, will be in higher demand in the future. D'Lima et al. [28], focusing on the housing market in the United States, find that demand for apartment properties, such as studios, will be impacted by both the preferences for more space as well as the need for improved technological infrastructure allowing for teleworking, thus producing decreases in housing prices in downtown areas, and premiums in the suburbs.

Duca et al. [27] argue that persisting house price indices do not sufficiently capture the change in economic preferences and that additional factors, such as behavioural changes, need to be accounted for. The study argues that the underlying income elasticity outlines a stronger preference for larger units and more space among wealthier households. Capturing these changes in economic preferences requires novel approaches and data sources, making use of big online data. In this regard, the Bank of Italy published a study on the change in preferences in housing demand using a multi-method approach, combining surveys and housing advertisements on online housing platforms, finding a particularly strong upward trend in the popularity of larger outdoor amenities [31]. Urban policy and planning can benefit from such novel approaches to studying changing preferences, as argued by [32], considering the allocation of retail, commercial and industrial properties.

Complementing these various insights from the literature, our study investigates shifts in Vienna's residential rental market in relation to the changes in housing preferences from 2018 to 2021/22, employing a hedonic price model to analyse premiums associated with housing attributes amidst rising telecommuting trends and inner-city transformation. We focus on the case study of Vienna as the city's layered urban structure with a historical core and a periphery mainly characterised by new buildings is particularly well suited to study urban landscapes in transition. We hypothesise that changed housing preferences are related to trends towards remote work and urban flexibility, driven by preferences for features like outdoor spaces. Conversely, attributes linked to public transit access are expected to diminish in importance, reflecting reduced reliance on traditional office commutes. Our analysis of 120,000 rental listings from a large Austrian online real estate platform suggests that landlords indeed seem to have adjusted their premiums for such features.

## Data and methods

### Collection of rental market data

In order to obtain the required data for our analysis, we extracted structured information and texts from Viennese rental apartment advertisements offered at the real estate listing online platform Willhaben over a period of six months in 2018 (23rd April 2018 to 7th November

2018) and during autumn and winter 2021/22 (15th November 2021 to 8th June 2022), see Fig 1A. In total, we collected data on 39,800 online listings in 2018. When we repeated the data collection 18 months after the onset of the Covid-19 pandemic, we collected data on another 81,900 apartments. The data collection worked as follows: every day at midnight, a web scraper would access the Willhaben website and collect the links to all individual apartments in Vienna listed on the platform. In a second step, the scraper would compare new links found on the website with a data base of already collected apartments. For new apartment listings, the scraper would then go to each individual link and collect the following elements: name, size, listing code, number of rooms, monthly rent, address (if available), the text description of the listing, a text element called 'info', location (mostly the district), a text field

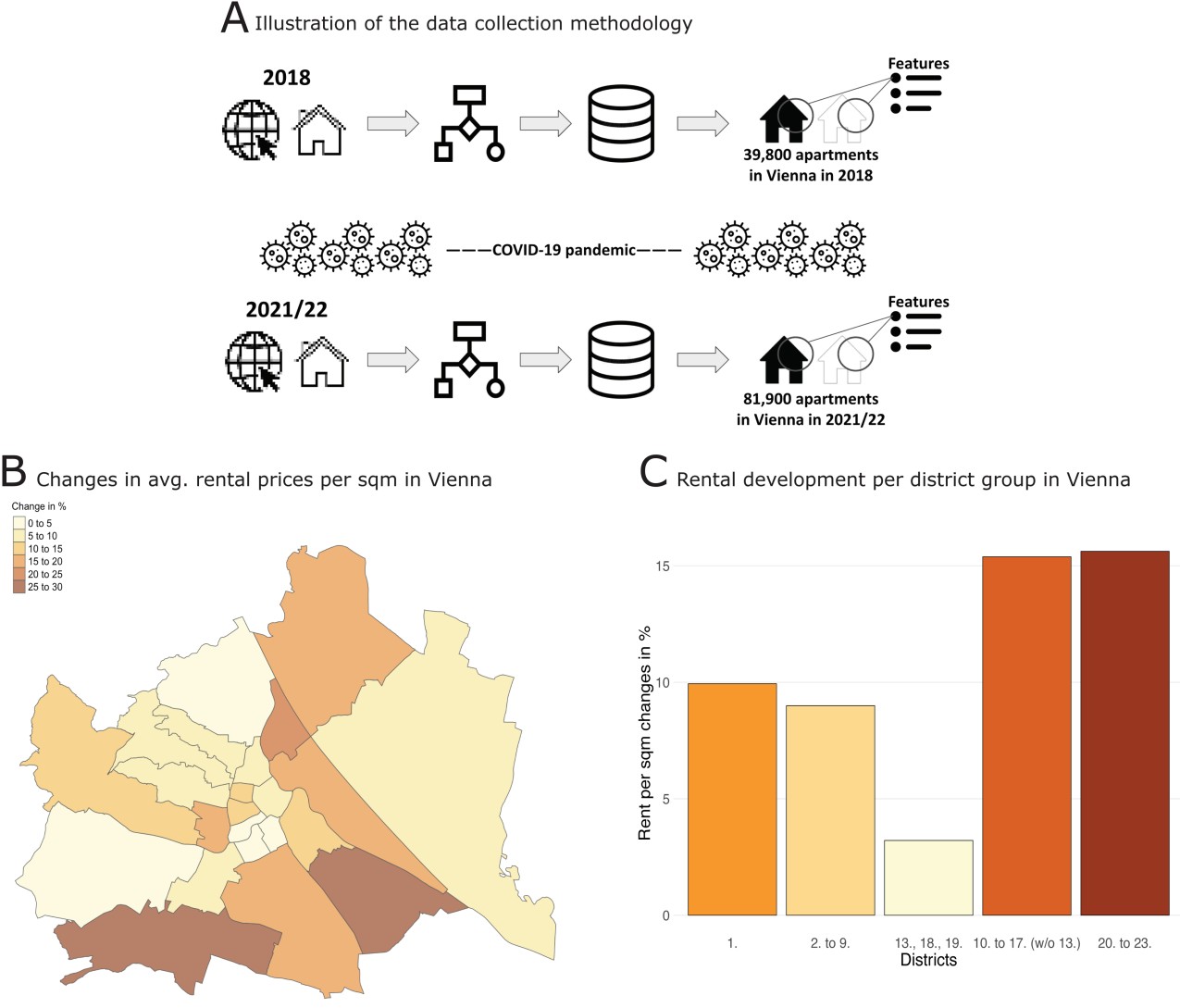

**Fig 1. Overview of data collection methodology and descriptive statistics. (A) Illustration of the data collection methodology: we collected data on approximately 40,000 apartments in 2018 and data on 82,000 apartments in 2021/22 from a large Austrian real estate listing online platform. (B) Changes in average monthly rent levels in Vienna over time: between 2018 and 2021/22, rent levels in Vienna increased on average by around 9 %, but there are substantial fluctuations between the districts. (C) Increases in rent levels were highest in the outer districts of the city.**

called 'equipment' (i. e. amenities), information about the heating, an ID of the broker (if that information was available), and information about the broker, if available.

This raw data contains a large variety of features per listing, not only the location (at the district level) but also monthly rent, basic hedonic features as well as the descriptive texts of the listings. The text description of the listings allowed us to extract further information about various amenities, such as the presence of outdoor areas, the connection to public and private transport or the heating system. To extract this information, we used a keyword analysis. In extracting meaningful keywords that would likely be relevant in the context of the Viennese rental market, we applied our own expert judgement and we collaborated with a local real estate broker who provided us with a list of keywords that would indicate relevant attributes of an apartment that could possible have an impact on the rent level. This way, we extracted a total of 41 binary variables capturing various features of each individual apartment as well as the building in which the apartment is located (further insights into the various apartment features in Fig 2 below).

To clean the data from potentially erroneous observations, we removed duplicates (i. e. apartments with the same ID), and we removed outliers such as apartments with reported sizes smaller than 10m$^2$, rent levels lower than € 99 per month or higher than € 19,000. Next to the 41 variables capturing amenities (type of heating, outdoor area, parking, etc.) we also extracted four identifiers (ID, URL, listing title, address if available) and basic hedonic features (location on the district level, monthly rent, size in square metres, number of rooms), resulting in a total of 56 features from the data.

Furthermore, the data was divided into five categories according to their location: the historic old town, the inner districts (2nd to 9th district), historic outer districts (13th, 18th and 19th district), and the outer districts in two groups: 11th to 17th district and 20th to 23rd district. This grouping covers the main differences in the structure of the city. The historical old town and the historical inner and outer city districts have a large share of old buildings, whereas the outer districts have a large share of newly built houses.

### Decomposing rent levels via hedonic price modelling

To analyse the data, we provide a number of descriptive statistics in addition to inferential and machine learning-based analyses. We explore machine learning tools to uncover non-linearities in the data: a decision tree algorithm (Fig 3A) visualises the complex interactions between the features in different districts, and a regularised model (*LASSO*, see [33]) reduces the feature space to the most influential variables relating rents and amenities (more details below and in Fig 3B). The main statistical tool being used to decompose rent levels is a hedonic price model.

Hedonic price models represent a widely used approach in real estate economics and housing markets [34–36]. These models build on the hedonic demand theory aiming to decompose the house price or monthly rent into its constituent characteristics, thereby revealing the implicit valuation of each attribute [37]. For instance, a house's price can be viewed as reflecting not only its physical features (e.g., size, number of bedrooms, location) but also its amenities (e.g., type of heating, proximity to public transportation) or other characteristics. Thanks to the granularity of data on amenities available in the non-structured text data contained in the online descriptions of the apartment listings, we can estimate the coefficients associated with many different attributes at the same time.

### Results

Rents in Vienna have not increased uniformly across districts between 2018 and 2021/22 (Fig 1B). Overall, rental prices rose by 9.2 % during this time, but the increases differed

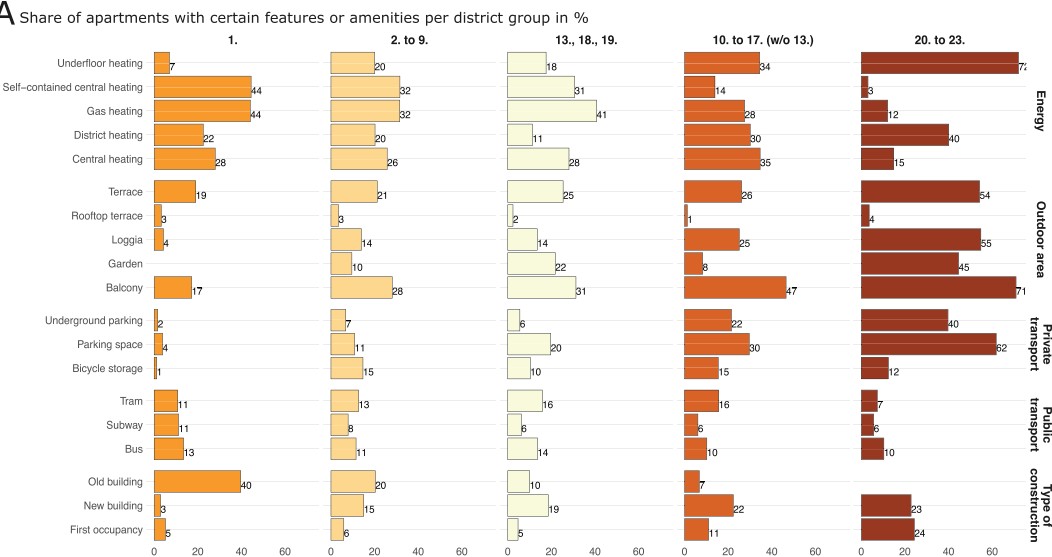

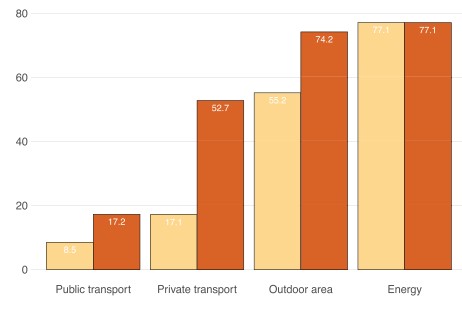

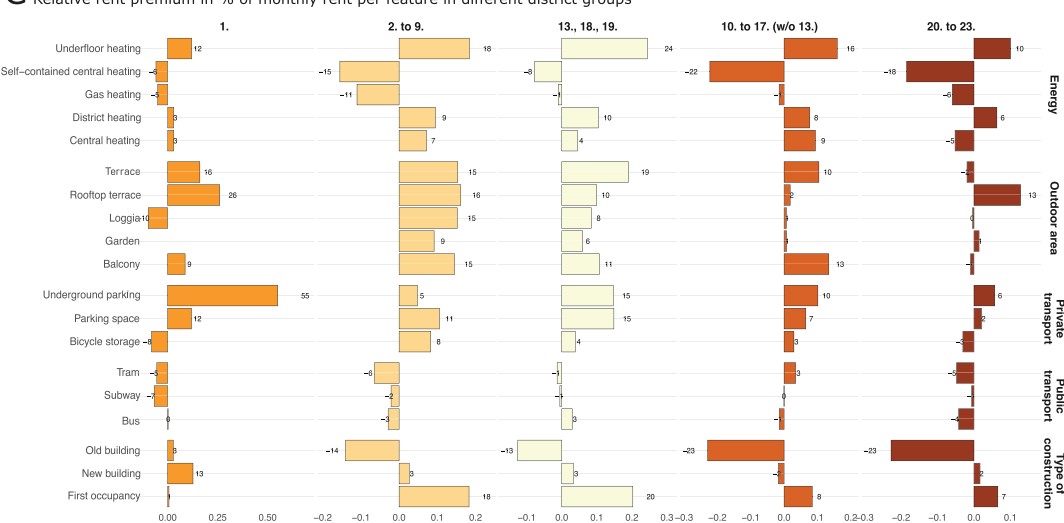

**Fig 2. Distribution of hedonic features and amenities by district and over time. (A)** Distribution of features and amenities in apartments in different districts of Vienna in 2021/22: in the outer districts (20th to 23rd district), the share of novel features such as underfloor heating is highest. **(B)** Development of features mentioned in listings over time: The share of 'private transport' and 'outdoor area' amenities grew considerably from 2018 to 2021/22. **(C)** Relative rent premium (% of the monthly rent) for features in different districts: scarce features such as rooftop terraces or underground parking come with considerable premium in the historic old town (1st district).

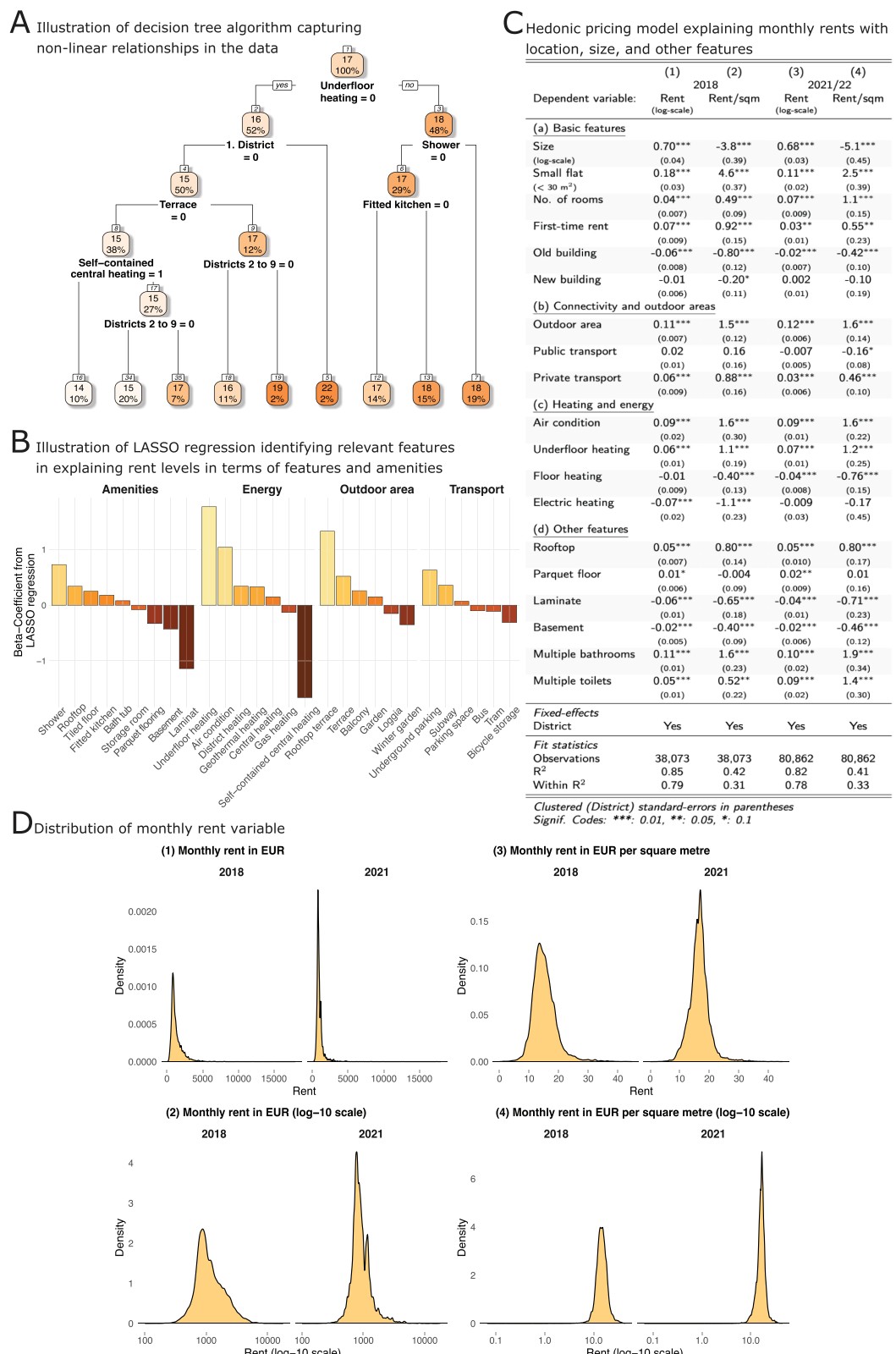

**Fig 3. Relating rent levels to hedonic features and amenities. (A) Illustration of a decision tree trained on the 2021/22 data: the model captures non-linear relationships relating basic features and amenities to rent levels. (B) Illustration of a regularised regression model (LASSO): the model identifies a smaller subset of influential variables associated with**

rent levels. **(C)** Regression models relating the rent level (€) (models 1 and 3, log-scale) and €per sqm (models 2 and 4) to (a) basic hedonic features, (b) connectivity and outdoor areas, (c) heating and energy, and (d) other features in apartments in Vienna in 2018 and 2021/22: besides basic hedonic features, amenities explain a large share of the overall variance. Comparing the coefficient estimates over time, a number of features that are potentially indicative of changed customer preferences related to the Covid-19 pandemic have changed in relevance: for example, the estimated rent effect of additional rooms has almost doubled; similarly, apartments with an outdoor area or multiple toilets obtain higher rents. On the other hand, small apartments have become less attractive. **(D)** Distribution of the untransformed and log-transformed dependent variable *monthly rent in EUR* (panels 1 and 2) and *monthly rent in EUR per square metre* (panels 3 and 4).

substantially between the districts. The first district had already charged the highest rents per square metre in 2018; the increases were close to the city-wide average. The same goes for the historical inner city districts (2nd to 9th district). In contrast, the outer districts (10th to 17th district as well as 20th to 23rd district) saw the highest increases with over 15 % on average (Fig 1C). The rent levels increased much slower in the historical outer city districts (13th, 18th, and 19th districts) with only 3.2% on average. Thus, the outer districts caught up somewhat with the higher rents in the city centre, levelling out the rental disparities between the inner and outer parts of the urban area.

By analysing the individual features and amenities of the apartments listed, we can understand how the structure of the city—shaped by the difference of historical neighbourhoods characterised by buildings from the late 19th and early 20th century in the city centre and newly-built neighbourhoods in the outer districts—essentially leads to distinct housing markets, which have been differently affected by the changing housing preferences as a reaction to the increased work-from-home routines caused by the pandemic. Fig 2 visualises the results of this analysis.

Fig 2A shows the distribution of a number of features and amenities (grouped into the categories 'Energy', 'Outdoor area', 'Private transport', 'Public transport', and 'Type of construction') in the five district groups. There are obvious differences between the historical and outer districts with regard to frequently mentioned features such as heating type, outdoor areas and parking spaces. The first district, for example, is characterised by gas heating and self-contained central heating, while the majority of the listed apartments in the outer districts offer new or more energy-efficient heating types, such as underfloor heating or district heating. Similarly, the share of apartments with an outdoor area, such as a terrace, loggia, or balcony, is much lower in the historical districts than in the outer districts. Comparing features related to transport and connectivity shows the biggest contrasts between both parts of the city: While the share of private transport-related facilities (underground parking spaces, parking spaces, bicycle storage) is substantially higher in the outer districts, the inner districts tend to be better connected to public transport. In interpreting the public transport connectivity, however, one has to acknowledge that Vienna is, overall, considered to have one of the best public transport systems globally, with very good and dense coverage in all parts of the city. This is probably a reason why proximity to public transport is not considered a distinguishing feature of apartments in the city. The category 'type of construction' underlines the structural differences between the Viennese districts: While 40 % of the apartment listings from the first district explicitly mention being in an old building, this term was very rarely, if ever, mentioned in the listings in the outer districts.

Comparing the share of listings mentioning particular features between 2018 and 2021/22 (Fig 2B), we see that some features have become more frequent (or were at least more frequently mentioned). Building features related to private transport have seen the biggest leap: The share of apartments that mention at least one private transport-related keyword tripled

from one in six apartments to more than one in two apartments between both observation periods. Similarly, the share of listings that mention public transport-related features or outdoor amenities also grew over time, but both to a smaller extent. Partially this can be attributed again to the rise in new buildings as building regulations impose one parking spot per every 100 square metres of living space in newly constructed flats in Vienna [38]. Additionally, parking as well as proximity to public transport are more relevant in the outer districts where new constructions are located. The share of apartments describing the type of heating did not change over time.

To elaborate further on the relationship between housing preferences and the spatial distribution of amenities in the different parts of the city, we analyse the rental premiums for individual features in the five district groups in Fig 2C by comparing apartments that list the respective features with those that do not include them. Each bar represents the rent premium in per cent of the monthly rent for apartments in a respective district group that lists the respective features vs those that do not. Similar to the top panel of Fig 2, we see differences between the historic districts and the outer districts. Here, however, the gap in rent premiums tends to be inversely related to the features' relative frequency or scarcity. For example, modern types of heating, such as underfloor heating, see a large positive rent premium in the historic districts, where these features are relatively scarce compared to the outer districts. Outdated types of heating, which are environmentally and financially costly, such as self-contained central heating, see negative rent premiums across the board.

The differences between historical and outer districts become more obvious when looking at outdoor area features: Rooftop terraces and other terraces, which are rare in the historical parts of the city, obtain a positive rent premium there, while this is not necessarily the case in the outer districts. The biggest rent premium, however, comes from underground parking facilities in the first district. In the inner city, parking spaces are very scarce and underground parking facilities are potentially the only reliable option, which is why apartments that have access to underground parking see substantially higher rents in this part of town. Rent premiums for this or related features are more moderate in the other inner districts and largely absent in the outer districts, where parking spaces are much more abundant. Echoing the relative unimportance of public transport as a distinguishing feature of apartments in Vienna, this feature type is not associated with higher rent levels in any part of the city. The value of different types of construction, however, is related to positive or negative rent premiums. Apartments in old buildings ('*Altbau*') are characterised by lower rent levels in most parts of the town, with negative rent premiums being larger in the outer districts where newly constructed buildings are more frequent; this mirrors the current rent control scheme in Vienna that is attached to building age. In the historical districts (2nd to 9th district and 13th, 18th, 19th district), newly refurbished apartments (first occupancy, '*Erstbezug*') see considerable positive rent premiums.

The relationship between rent levels, basic hedonic features and amenities is further analysed in Fig 3. In panel 3A, we illustrate the partly non-linear associations in the data by visualising a decision tree. It aims to classify apartments according to rent levels by selecting predictors and cut-off values. The decision tree algorithm starts at the top by identifying underfloor heating to be a main distinguishing factor — even more important than location — in determining rent levels. Other important features in this run of the algorithm are outdoor space (terrace), heating and kitchen type, location in the historic inner city and apartments that explicitly mention showers as a bathroom feature. To illustrate the findings, apartments without underfloor heating (node 1: left) and in the first district (node 2: right) obtain the highest

average rent levels with € 22 per square metre. Apartments without underfloor heating, outside of the first district, without a terrace and with self-contained central (gas) heating yield the lowest average rents with € 14 per square metre.

Note that the visualisation displayed here serves only as an illustration to reveal the partly complex correlational structures in the data. It is the result of only one run of a decision tree model with a 'complexity parameter' of 5 % (i. e., splits are only conducted if the gain in $R^2$ is at least 5 %). Different results would have been obtained if other pruning or averaging strategies had been applied.

Fig 3B provides another perspective on the correlations between apartment features and rent levels. The figure visualises the coefficient estimates of a regularised LASSO regression model, including all available features (as z-scores, i. e., standardised to have a mean of zero and a variance of one) and rent levels as a bar chart. The LASSO model shown here serves the purpose to explore the data set for potentially influential factors among the vast number of variables in our data set. It helps to guide the feature selection for the hedonic pricing model. Intuitively, the LASSO method acts like a built-in feature selector: it adds a penalty parameter to the minimisation problem solved by the regression model. The penalty forces parameters to be dropped from the model if they add little to its explanatory power, striking a balance between complexity and accuracy. In applying a shrinkage parameter to the model, LASSO effectively selects only those variables that contribute to explaining the variance substantially, and it shrinks the coefficient estimates of other, less relevant variables to zero. As a result, not all features are included in the model displayed in Figure B. Corroborating the findings from Fig 3A, the model identifies underfloor heating, self-contained central heating and rooftop terraces as important determinants of rent levels. In addition, the model finds the presence of air conditioning ('*Klimatisiert*'), underground parking and laminate to be relevant features.

Complementing the results presented so far, Fig 3C shows four regression models that relate the total rent (models 1 and 3) and rent per square metre (models 2 and 4) in 2018 and 2021/22 to various features categorised in four groups: basic hedonic features, connectivity and outdoor areas, heating and energy, as well as other features. We have log-transformed the dependent variable *monthly rent in EUR* in models 1 and 3 to account for the skewed distribution of the variable (see Fig 3D). In the feature selection for the regression models, we have used information from the analyses shown in Fig 1 to Fig 3B to guide the inclusion of variables and applied stepwise variable selection, aiming to include only those factors that have been identified to be statistically significant in at least one of the regression models. All models include district-level fixed effects and the standard errors are clustered at the district level to account for possible common unobserved influential factors among apartments in the same district.

Overall, we find that both basic features and amenities influence rent levels, largely in line with our research hypothesis and the descriptive statistics reported in this section. Smaller apartments with several rooms obtain high rents per square metre, and newly refurbished apartments obtain a positive rent premium. In both observation periods, outdoor areas increase the absolute and per square metre rent, as does air conditioning, private transport facilities, underfloor heating, a rooftop apartment, and multiple bathrooms and toilets.

Some of the features that are relevant to re-purpose apartments as places of living and working, such as multiple rooms to establish a dedicated home office, an outdoor area for recreation, as well as multiple toilets for co-use, all saw their rent premiums increase both in absolute and per square metre terms.

Other features that are associated with non-significant or negative rent premiums are the type of construction itself (old building), floor heating, the type of flooring (laminate) and the presence of a basement. The negative rent premiums for certain features (old building,

basement) are likely explained by other features capturing the association with rent levels, leaving a certain sub-group of the data set being characterised by these features but lacking other amenities that have a positive effect on rents. The models explain a substantial part of the overall variation in the data with $R^2$-values between 41 % and 85 %.

## Discussion

### Summary of the main findings

Our study shows that features of residential properties that allow the multiple use of apartments as places of living and working are associated with higher rent premiums in 2021/22 compared to 2018. While our correlational research design does not allow causal claims, the differences observed over time are (at least to some extent) likely to be related to changing housing preferences. Price signals in the Viennese housing market translate these preferences into market outcomes. The relative scarcity of popular features such as underfloor heating, underground car parks or outdoor spaces is reflected by higher rent premiums, particularly in those parts of the city where these features are rare. Features associated with an outdated fit-out, especially those that are not climate-friendly, such as self-contained central (gas) heating, come with a rental discount.

The influence of changed housing preferences is particularly pronounced when comparing the historical inner city of Vienna (1st district) with the outer districts: Space and green areas are in short supply in the first district, while the outer districts are characterised by more space and their proximity to surrounding suburban and rural areas. There are more old buildings in the inner district (with outdated fit-outs), while new buildings are constructed in the outer districts to meet updated housing preferences. The diverging rent premiums in these two parts of town can be attributed to these differences in the housing stocks: While the potential for rent increases in the existing stock is limited because the flats do not change much in terms of their quality (and most of them are rent-controlled), newly-built projects can cater to changing customer preferences. New buildings in the outer districts often feature modern amenities such as underfloor heating or parking spaces, but as they are relatively abundant there, they will come only with a moderate rent premium. In contrast, the same features will have much higher premiums in the historical inner part of the city because of their scarcity here.

The machine learning models and the hedonic pricing model confirm the descriptive statistics. They reveal that quality-enhancing features that meet changing housing preferences are associated with higher rent levels. The features that saw their rent premiums increase are those that are in line with the changing living and working habits associated with the hybrid and remote work patterns induced by the Covid-19 pandemic.

### Other factors influencing the observed trends in the data

The pandemic was not the only change in the macro-conditions that might have had an impact on the rental market in Vienna. Here, we are discussing a number of other contextual factors that changed between the two observation periods in 2018 and 2021/22.

Rental prices during the observation period were partially fuelled by low interest rates. In response to the pandemic crisis, the ECB replied in an even more expansionary manner that lead to a further decrease in mortgage rates from 1.83 per cent in 2018 to 1.2 per cent in 2021 [39]. To understand the impact of the monetary policy on the housing market, [40], for example, analyse 35 million real estate listings in Germany between 2007 and 2023. They

find that expansionary monetary policy increased housing prices and rents. [41] find similar results for a panel of European countries between 2010 and 2019.

Furthermore, the completion of new dwellings in Vienna increased from an average of 6,500 per year between 2005 and 2017 to 13,600 in 2018 and 16,200 in 2022 [42]. With increasing construction cost [43], rents in newly-built dwellings are typically above the existing average [44]. [45] proposes a model to understand to what extend house prices are driven by fundamentals for the case study of Vienna. While construction costs do play an important role, the findings suggest that the price development in Vienna exceeds the fundamentals. However, the geographical location of the new constructions, as well as construction cost development could explain part of the convergence in rent levels between the historical inner district and the outer districts, where most of the newly-built dwellings are located [46].

In light of the start of the war in Ukraine and the upcoming energy crisis in Austria, energy efficiency, as well as the energy source, might play a more important role in 2022 compared to 2018. While many energy contracts have a fixed rate over a year, a contract for new tenants would have increased substantially, with the energy-price-index increasing by 32 per cent between September 2021 and March 2022 while the overall consumer price index increased by five per cent [47].

## Implications for theory and practice

The changed work and living arrangements in urban areas, fuelled by the Covid-19 pandemic, underscore the imperative for policymakers to adapt strategies that foster resilient and inclusive urban environments [1], which have been identified by real estate associations and investors [48–53]. A salient feature is the progressing hollowing-out of central business districts as traditional work patterns shift [4]. To react to this trend, policymakers should prioritise initiatives aimed at diversifying economic activities and cultivating vibrant community spaces within central business districts but also aim at providing urban infrastructures in the outer parts of urban areas, as these are places of increased activity in the hybrid world of work.

As the nature of work transforms, residential infrastructure must evolve in tandem to accommodate these novel trends [1]. While it is unlikely that employers will rely on fully remote employment at a large scale [54], the spread of hybrid employment forms is paradigmatic. Apartment design should react to this by prioritising flexibility and functionality, providing residents with adaptable spaces that allow them to work remotely and cater to their personal well-being. At the same time, urban mobility patterns are changing: To reduce the reliance on centralised transit systems, investments in on-demand forms of public transportation and improved urban connectivity via bike lanes, as well as bike-, scooter- and car-sharing, are required to support novel and more decentralised commuting patterns.

The proliferation of remote and hybrid forms of work also underlines the critical importance of high-speed internet access in residential areas. Expanding internet infrastructure in these areas will ensure equitable access across diverse communities, bridging existing digital divides and facilitating greater digital inclusion and business opportunities. Additionally, the redesign of office spaces is imperative to create engaging and collaborative environments that attract employees back to physical workspaces, thereby stimulating productivity, well-being, and innovation.

In order to manage the required changes to the urban infrastructure, informed decision-making hinges on the effective utilisation of digital data to understand evolving consumer preferences and behaviours in real time. Our study shows that integrating online data collection helps to nowcast and anticipate market trends, which helps to tailor interventions to

meet evolving needs more effectively. Having access to reliable and up-to-date data is particularly relevant in capital-intensive and slow-moving sectors of the economy, such as real estate, as these data can help investors and other stakeholders to closely follow evolving trends and adjust investment, planning and construction strategies [48,55,56]. Moreover, the evolving nature of work-from-home practices also necessitates a reevaluation of social housing policies, emphasising the provision of places suitable for remote work in those parts of the housing market that focus mostly on affordability. This is important to mitigate disparities and promote social mobility.

## Methodological limitations

As our study relies on the analysis of online data, potential sample biases could occur, as the demographics of online users may not be representative of the broader population, for example, by disregarding social housing, which tends not to be listed online. Furthermore, the reliance on rent information sourced from online platforms introduces a degree of uncertainty, as ask prices may not necessarily reflect final transaction rents agreed in the tenancy contract.

Moreover, the process of data collection from online platforms is contingent upon the algorithms utilised by the respective companies, which may inadvertently introduce biases in the representation of available apartments. We collected data during two observation periods in 2018 (23rd April 2018 to 7th November 2018) and 2021/22 (15th November 2021 to 8th June 2022). We cannot control for potential changes to the website design, which might have an influence on the display of keywords or other unstructured data collected from the descriptions of the online listings. Additionally, seasonal effects could also have had some effect, but due to the overall large sample size and partly overlapping data collection periods in each calendar year, we do not assume seasonalities to have played a big role in contrast to the large change that was brought about to the rental market by the pandemic.

The statistical findings reported in this study are correlational: The research design employed here does not allow us to make causal claims or run quasi-experimental analyses, and we cannot control for potential other factors that might influence changing housing patterns, such as increased environmental awareness. However, this unobserved heterogeneity does not affect the main argumentation of this study, namely that online data can help to track and understand (rapidly) changing housing preferences, such as in times of the Covid-19 pandemic, which will play out differently in cities based on their urban structure and connectivity.

Despite these methodological limitations, the availability and size of online data justifies its utilisation for nowcasting and providing timely insights into housing market trends. The unparalleled granularity and timeliness offered by online datasets present valuable opportunities for augmenting traditional survey-based and registry-based approaches, albeit with due diligence to mitigate potential biases and uncertainties inherent in online data sources.

## Conclusion

We have investigated how changing housing preferences – induced mainly but not entirely by the Covid-19 pandemic – are reflected in rent levels and premiums for certain housing features. We have used online real estate data to track such changes in the city of Vienna. In total, we investigate a data set of 120,000 apartments collected from a large Austrian real estate platform in 2018 and 2021/22.

We find that rents have developed unequally in the city. The outer districts have caught up; ask rents in districts with large shares of newly-built homes have increased much stronger

between 2018 and 2021/22 than in the historic districts. Both parts of the city are pretty distinct in terms of the housing they have to offer: The historical city centre consists of old buildings with, e. g., outdated heating systems, few parking opportunities but very good access to public transport and the city centre, while the outer districts provide new homes with modern fit-outs, access to local recreation areas but longer commutes to the inner city.

It seems that landlords anticipate housing preferences have come to favour the outer districts more heavily; a part of this should be due to the trend towards work-from-home schedules triggered by the Covid-19 pandemic. Tenants seem no longer to attach much value to the locational advantages of the inner city. Using machine learning and hedonic pricing models, we find that rent premiums for a number of features have changed between 2018 and 2021/22. New and spacious apartments with several rooms now come with increased rent premiums. The same is true for apartments with outdoor areas or multiple bathrooms. On the other hand, small apartments that do not allow working from home have become less attractive. Connectivity to public transport has even switched signs. Not only does it seem to be no longer a feature associated with higher rent levels, but it also comes with a negative premium.

Our analysis underlines the importance of real-time data in predicting shifts in consumer preferences in a timely manner. This holds especially for slow-moving sectors like the real estate industry, where adaptations to rapid changes in demand patterns—like the ones triggered or amplified by Covid-19—take a long time to implement.

## Acknowledgments

We are incredibly grateful to Fabian Stephany and Moritz Marpe for their support and help on earlier versions of the manuscript. In addition, we would like to thank the whole team of the Future of Real Estate Initiative at the Saïd Business School—in particular Andrew Baum and Andy Saull—for helpful comments on the research project, as well as the participants of the PropTech Vienna Conference 2023. This paper is the result of a commissioned research project by Agenda Austria to the DWG Data Science Company.

## Author contributions

**Conceptualization:** Fabian Braesemann, Jan Kluge, Hanno Lorenz.

**Data curation:** Fabian Braesemann.

**Formal analysis:** Fabian Braesemann, Jan Kluge.

**Funding acquisition:** Hanno Lorenz.

**Investigation:** Fabian Braesemann, Jan Kluge, Hanno Lorenz.

**Methodology:** Fabian Braesemann.

**Project administration:** Fabian Braesemann.

**Resources:** Fabian Braesemann.

**Software:** Fabian Braesemann.

**Validation:** Fabian Braesemann.

**Visualization:** Fabian Braesemann.

**Writing – original draft:** Fabian Braesemann, Jan Kluge.

**Writing – review & editing:** Fabian Braesemann, Jan Kluge.

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
