## [Decision Letter · Decision Letter 0]

4 Nov 2024

PONE-D-24-30670How have urban housing preferences developed in response to the Covid-19 pandemic?PLOS ONE

Dear Dr. Braesemann,

Thank you for submitting your manuscript to PLOS ONE. After careful consideration, we feel that it has merit but does not fully meet PLOS ONE’s publication criteria as it currently stands. Therefore, we invite you to submit a revised version of the manuscript that addresses the points raised during the review process. **In particular, I encourage the authors to address the issues raised by Reviewer 2 regarding the methodologies and potential confounding factors for the analysis.**

We look forward to receiving your revised manuscript.

Kind regards,

Maurizio Fiaschetti

Academic Editor

PLOS ONE

**Journal Requirements:**

4. We note that Figure 1B in your submission contain map images which may be copyrighted. All PLOS content is published under the Creative Commons Attribution License (CC BY 4.0), which means that the manuscript, images, and Supporting Information files will be freely available online, and any third party is permitted to access, download, copy, distribute, and use these materials in any way, even commercially, with proper attribution. For these reasons, we cannot publish previously copyrighted maps or satellite images created using proprietary data, such as Google software (Google Maps, Street View, and Earth). For more information, see our copyright guidelines: http://journals.plos.org/plosone/s/licenses-and-copyright.

We require you to either present written permission from the copyright holder to publish these figures specifically under the CC BY 4.0 license, or remove the figures from your submission:

a. You may seek permission from the original copyright holder of Figure 1B to publish the content specifically under the CC BY 4.0 license.  

Reviewers' comments:

Reviewer's Responses to Questions

**Comments to the Author**

1. Is the manuscript technically sound, and do the data support the conclusions?

Reviewer #1: Yes

Reviewer #2: No

2. Has the statistical analysis been performed appropriately and rigorously? 

Reviewer #1: Yes

Reviewer #2: No

3. Have the authors made all data underlying the findings in their manuscript fully available?

Reviewer #1: No

Reviewer #2: No

4. Is the manuscript presented in an intelligible fashion and written in standard English?

Reviewer #1: Yes

Reviewer #2: Yes

5. Review Comments to the Author

**Reviewer #1: **The research article titled “How have urban housing preferences developed in response to the Covid-19 pandemic?” aims at exploring how the Covid-19 pandemic have influenced housing preferences in Vienna. By analysing a large database of apartment listings from an online real estate platform, the authors identify changes in rent premiums for various amenities and features, highlighting a shift towards preferences in the analysed period.

The study addresses a highly relevant topic, that fits in the current literature about the impacts of the Covid-19 pandemic on urban living and working conditions. The literature review on the relationship between the changes induced by the pandemic and the housing prices is well written and complete. The analysis is rigorous, and the figures are clear. Moreover, the authors clearly state the limitations of their analysis, highlighting the possible bias in the data collection, as well as the lack of causal claims of the results.

However, I have some concerns:

DATA

The data source, such as the procedure followed to obtain the analysed data is not fully clear. The authors should deeply describe the text-mining procedure followed to get the data, as well as the data cleaning procedure. Without a clear description of the procedures, it is difficult to assess the reliability of the data.

RESULTS

It is necessary deep the reasons why in the period 2021/2022 the availability of transport network leads to a reduction of the rent (significant and negative effect). This result is unexpected and should be carefully interpreted, otherwise the entire analysis loses credibility.

OTHER

- Even if the authors deeply describe the results with extensively description of Vienna’s housing dynamics and specific characteristics, the results seem quite city-specific. The focus on a specific case study (together with the lack of a causal claim) limits the external validity of the study. For this reason, I would suggest to partially change the title of the research mentioning the name of the city.

- At page 17 the authors write that “we have used information from the previous analyses to guide the inclusion of variables”. It is necessary to cite the articles (at least some of them).

**Reviewer #2:** • Data Clarity and Representativeness:

The data presented in the paper is insufficiently described. There is too little information about the company providing the data, and no discussion regarding its representativeness. Additionally, there is a lack of detail about how the data is structured and how it was obtained, aside from a brief mention that it was scraped and abstracted using text recognition. Crucial details, such as the precise time periods during which the data was collected, are also missing. The paper mentions that data was collected over six months in 2018 and during fall/winter 2021/2022, but it is unclear whether this also covers a six-month span. This lack of clarity on the temporal aspect of the data collection raises concerns, as differences in the time periods may explain observed variations, potentially independent of the pandemic's effects, such as seasonal effects.

• Contextual Factors Beyond the Pandemic:

The paper does not sufficiently discuss other significant events that occurred during the studied period, aside from the pandemic, which could have affected the results. For instance, the Russian invasion of Ukraine led to sanctions that impacted gas prices. This may explain the increase in premiums for floor heating, particularly in regions like Austria, where, to my knowledge, Russian gas imports continued. A discussion of these additional factors is necessary to avoid attributing all observed changes solely to the pandemic.

• Statistical Models and Methodology:

The statistical models, particularly the LASSO model and conventional regression models, are poorly described. A more detailed and intuitive discussion of the properties of these models is required. Furthermore, the paper lacks essential information regarding model implementation, such as whether standard errors were clustered, which is a critical omission.

• Lack of Descriptive Statistics:

The absence of descriptive statistics for the data severely limits the ability to assess the models' reasonableness. For example, there is no information on the distributional properties of key variables, such as rent, which makes it impossible to determine whether a level or log transformation would be more appropriate for the dependent variable. Similarly, without details about the explanatory variables, it is difficult to evaluate the relevance and reliability of the results. Providing descriptive statistics and an appropriate discussion of the data’s properties is essential for a more thorough assessment of the models and findings.

6. PLOS authors have the option to publish the peer review history of their article (what does this mean?). If published, this will include your full peer review and any attached files.

Reviewer #1: No

Reviewer #2: No

---

## [Author Response · Author response to Decision Letter 1]

26 Feb 2025

Please find the response to the reviewers in the letter attached to this submission.

---

## [Decision Letter · Decision Letter 1]

25 Mar 2025

How have urban housing preferences developed in response to the Covid-19 pandemic? A case study of Vienna

PONE-D-24-30670R1

Dear Dr. Braesemann,

We’re pleased to inform you that your manuscript has been judged scientifically suitable for publication and will be formally accepted for publication once it meets all outstanding technical requirements.

Kind regards,

Eda Ustaoglu, PhD

Academic Editor

PLOS ONE

Additional Editor Comments (optional):

Reviewers' comments:

Reviewer's Responses to Questions

**Comments to the Author**

1. If the authors have adequately addressed your comments raised in a previous round of review and you feel that this manuscript is now acceptable for publication, you may indicate that here to bypass the “Comments to the Author” section, enter your conflict of interest statement in the “Confidential to Editor” section, and submit your "Accept" recommendation.

Reviewer #1: All comments have been addressed

Reviewer #2: All comments have been addressed

2. Is the manuscript technically sound, and do the data support the conclusions?

Reviewer #1: Yes

Reviewer #2: Yes

3. Has the statistical analysis been performed appropriately and rigorously? 

Reviewer #1: Yes

Reviewer #2: Yes

4. Have the authors made all data underlying the findings in their manuscript fully available?

Reviewer #1: Yes

Reviewer #2: Yes

5. Is the manuscript presented in an intelligible fashion and written in standard English?

Reviewer #1: Yes

Reviewer #2: Yes

6. Review Comments to the Author

Reviewer #1: (No Response)

Reviewer #2: I have now read the revised version of manuscript. I find the revision carefully executed and I am satisfied with the handling of my main comments. I also appreciate the author’s responsiveness to the many additional constructive suggestions and comments in the review process: the new analyses are carried out in a satisfying way and in instances where no explicit action has been possible the author provides sufficient motivations. In sum, I am satisfied and advise that, what I now consider a valuable paper is published.

7. PLOS authors have the option to publish the peer review history of their article (what does this mean?). If published, this will include your full peer review and any attached files.

Reviewer #1: No

Reviewer #2: No

---

## [Editor Report · Acceptance letter]

PONE-D-24-30670R1

PLOS ONE

Dear Dr. Braesemann,

I'm pleased to inform you that your manuscript has been deemed suitable for publication in PLOS ONE. Congratulations! Your manuscript is now being handed over to our production team.

Kind regards,

on behalf of

Dr. Eda Ustaoglu

Academic Editor

PLOS ONE